# Transcriptional Differences in Lipid-Metabolizing Enzymes in Murine Sebocytes Derived from Sebaceous Glands of the Skin and Preputial Glands

**DOI:** 10.3390/ijms222111631

**Published:** 2021-10-27

**Authors:** Katharina Klas, Dragan Copic, Martin Direder, Maria Laggner, Patricia Sandee Prucksamas, Florian Gruber, Hendrik Jan Ankersmit, Michael Mildner

**Affiliations:** 1Laboratory for Cardiac and Thoracic Diagnosis, Regeneration and Applied Immunology, Department of Thoracic Surgery, Medical University of Vienna, 1090 Vienna, Austria; katharina.klas@meduniwien.ac.at (K.K.); dragan.copic@meduniwien.ac.at (D.C.); martin.direder@meduniwien.ac.at (M.D.); maria.laggner@meduniwien.ac.at (M.L.); hendrik.ankersmit@meduniwien.ac.at (H.J.A.); 2Aposcience AG, 1200 Vienna, Austria; 3Department of Dermatology, Medical University of Vienna, 1090 Vienna, Austria; 19imc11078@fh-krems.ac.at (P.S.P.); florian.gruber@meduniwien.ac.at (F.G.); 4Department of Medical and Pharmaceutical Biotechnology, University of Applied Sciences, 3500 Krems, Austria

**Keywords:** skin, sebaceous gland, preputial gland, scRNA-sequencing, lipid, sebocyte differentiation, bioactive lipid synthesis

## Abstract

Sebaceous glands are adnexal structures, which critically contribute to skin homeostasis and the establishment of a functional epidermal barrier. Sebocytes, the main cell population found within the sebaceous glands, are highly specialized lipid-producing cells. Sebaceous gland-resembling tissue structures are also found in male rodents in the form of preputial glands. Similar to sebaceous glands, they are composed of lipid-specialized sebocytes. Due to a lack of adequate organ culture models for skin sebaceous glands and the fact that preputial glands are much larger and easier to handle, previous studies used preputial glands as a model for skin sebaceous glands. Here, we compared both types of sebocytes, using a single-cell RNA sequencing approach, to unravel potential similarities and differences between the two sebocyte populations. In spite of common gene expression patterns due to general lipid-producing properties, we found significant differences in the expression levels of genes encoding enzymes involved in the biogenesis of specialized lipid classes. Specifically, genes critically involved in the mevalonate pathway, including squalene synthase, as well as the sphingolipid salvage pathway, such as ceramide synthase, (acid) sphingomyelinase or acid and alkaline ceramidases, were significantly less expressed by preputial gland sebocytes. Together, our data revealed tissue-specific sebocyte populations, indicating major developmental, functional as well as biosynthetic differences between both glands. The use of preputial glands as a surrogate model to study skin sebaceous glands is therefore limited, and major differences between both glands need to be carefully considered before planning an experiment.

## 1. Introduction

The skin provides a variety of adnexal structures, including the pilosebaceous unit, which comprises the hair follicle, arrector pili muscle and the sebaceous gland. The latter contributes critically to epidermal homeostasis and barrier function by secreting lipids and enzymes [1]. Sebaceous gland (SG) development is tightly coordinated with the formation of the hair follicle (HF) [2,3]. The most prominent functions of SGs are the production of sebum and the trafficking of lipids and lipid-soluble factors to the skin surface [4]. Sebum mainly consists of non-polar lipids, including triglycerides, fatty acids, wax esters, cholesterol and squalene [5]. Sebum production follows a highly specific program where pre-sebocytes undergo a maturation and differentiation process. During this differentiation program, sebocytes (SEB) from the outer epithelial layer of the sac-like gland structure move towards the lumen to the hair shaft, ultimately leading to cell death. The amount of intracellular lipids constantly increases during differentiation until the so-called “necrotic zone” is reached and terminally differentiated sebocytes undergo membrane lysis and nuclear degradation, thereby releasing their cellular content [5,6,7]. SG lineage differentiation relies on a well-balanced equilibrium of canonical Wnt/β-catenin and c-myc/Hedgehog signaling. SEB progenitors usually express high levels of β-catenin. Lymphoid enhancer-binding factor-1 (*Lef1*) serves as an essential regulatory factor controlling lineage differentiation towards either HF or SG cells [8,9,10]. Terminally differentiated SEB express high levels of stearoyl-CoA desaturase 1 (*Scd1*) and peroxisome proliferator-activated receptor γ (*Pparg*). Both factors are known to critically contribute to SG differentiation and the formation of a functional epidermal barrier by regulating keratinocyte differentiation [10,11]. SG function and lipid composition were linked to several inflammatory dermatoses, including, for example, acne vulgaris, atopic dermatitis, psoriasis, rosacea and seborrheic dermatitis with tremendously changed levels of squalene and bioactive sphingolipids such as ceramide (CER) [12,13,14,15,16,17,18,19,20,21].

Squalene represents one of the most abundant lipid components found in human sebum [22]. It serves as a precursor for cholesterol, which is lastly synthesized by epidermal keratinocytes rather than SEB, as they lack the necessary enzymes required for the final conversion steps [12,23,24]. Squalene synthesis occurs via the mevalonate/isoprenoid pathway, which yields either non-sterol compounds, such as ubiquitin, heme and dichols, or sterol compounds, including squalene and cholesterol.The mevalonate pathway is initiated by the conversion of acetoacetyl-CoA to β-hydroxy-β-methylglutaryl-CoA (HMG-CoA) by HMG-CoA synthase (*Hmgcs1*). The name-giving intermediate mevalonate is further synthesized from HMG-CoA by HMG-CoA reductase (*Hmgcr*). Upon several intermediate synthesis steps, mevalonate is converted to geranyl diphosphate, which serves as a substrate for farnesyl diphosphate synthase (*Fdps*). Upon synthesis of farnesyl diphosphate, the mevalonate pathway may follow either the non-sterol branch or the sterol branch. Within the latter route, squalene synthase (*Fdft1*) synthesizes squalene from its precursor farnesyl diphosphate, thereby catalyzing the first step of the cholesterol-specific sterol branch of the isoprenoid biosynthetic pathway [25,26,27,28,29].

Biologically active sphingolipids, including sphingomyelin, ceramide and sphingosine 1-phosphate, received increasing attention over the past years with regard to their critical role in signaling cascades, regulating a vast array of cellular functions [30]. Furthermore, these lipids are important for the establishment of a functional skin barrier [17,18,30,31,32,33,34,35]. Under homeostatic conditions, these lipids are formed via the sphingomyelinase pathway by de novo synthesis or by the recycling of sphingosine via the salvage pathway [36,37,38]. Depending on external stimuli, this sphingolipid synthesis favors different lipid products. UV-radiation, chemotherapy and death receptor ligation trigger CERs and sphingosine synthesis, functioning as potent pro-apoptotic mediators [30,39]. In contrast, growth factors (e.g., platelet-derived growth factor, insulin-like growth factor, vascular endothelial growth factor), cytokines such as TNF and interleukin-1, hypoxia and immune complexes promote sphingosine 1-phosphate synthesis [30]. In order to counteract the pro-apoptotic functions of CER, cells can evade CER-induced cell death by its conversion to non-apoptotic metabolites such as sphingosine 1-phosphate [40]. Sphingosine 1-phosphate and its synthesizing enzyme sphingosine kinase 1 were not only proven to promote cell survival and proliferation but also seem capable of actively inhibiting CER-mediated apoptosis [41]. The initial step of the de novo sphingolipid synthesis pathway occurs in the endoplasmic reticulum where serine palmitoyltransferase (*Sptlc1*) condensates serine and palmitoyl-CoA [36]. After several consecutive enzymatic intermediate steps, CER is formed, which can be further trafficked to the Golgi apparatus where it may be converted into sphingomyelin by sphingomyelin synthases (*Sgms1*) or into glucosylceramide by glucosylceramide synthase (*Ugcg*) and subsequently to glycosphingolipids [37,42]. The sphingomyelinase pathway is initiated with the hydrolysis of sphingomyelin by sphingomyelinases (SMases; *Smpd1*), resulting in CER generation [37]. CER is subsequently converted into sphingosine by pH-dependent ceramidases, including the alkaline ceramidase *Acer1*, neutral ceramidase *Asah2* and acid ceramidase *Asah1* [37,43,44]. Sphingosine is further processed to sphingosine-1-P by a sphingosine kinase (*Sphk1*) [38]. In the salvage pathway, recycling of pre-formed sphingolipids via ceramide synthase (*Cers4*) results in the degradation of sphingosine to CER [37,38].

Although there are advances in the in vitro approaches using sebocyte cell lines, such as SZ95, Seb-E6E7 and SEB-1, the major drawback of SG studies is the lack of adequate organ culture models, which are indispensable to comprehensively investigate sebaceous cell differentiation, lipid production and extrusion [45,46,47]. Since preputial glands (PGs) are significantly larger and easier to obtain, they were previously used as a surrogate model for skin SGs [7,48,49]. PGs are specialized sebaceous-like glands found in male rodents. Several studies investigating skin sebocytes were built on data obtained from PG sebocytes [50,51,52,53,54]. Here, we investigated the differences between SG- and PG-derived sebocytes on a single-cell level to elaborate their suitability as a model for skin sebaceous glands.

## 2. Results

### 2.1. Significant Differences in Gene Expression between Sebocytes Derived from Skin SGs and PGs

Since PGs were previously used as a model for skin SGs, we wanted to investigate their differences and similarities in more detail. Hematoxylin and Eosin (H&E) staining (Figure 1A) and immunofluorescence (IF) (Figure 1B) were performed to obtain an overview of potential morphological differences between SEB derived from skin (sSEB) and PG (pSEB) (Figure 1A,B). Both H&E staining and IF for Stearoyl-CoA desaturase (*Scd1*) revealed high morphological similarities between SEB of the glands of both tissue types. Skin sebocytes were only detected within the sebaceous gland in close proximity to a hair shaft. Preputial gland sebocytes were extensively distributed throughout the gland. Despite the larger size and therewith resulting higher numbers of SEB within the PG, overall morphology highly resembles sSEB. In both tissues, the gradual differentiation of SEB from less differentiated, nucleated SEB on the outer edges of the SG and further away from the lumen of the PG was observed. In SG as well as PG, nuclear degradation was detected in SEB closer to the hair canal or the glandular lumen, respectively, indicating a mature or fully differentiated state of the SEB. The size and gross shape of SEB were similar in both tissues.

In order to decipher transcriptional differences between sSEB and pSEB, we performed scRNAseq. Unbiased analysis of the data set and cluster generation revealed comparable cell type composition in both samples (Figure 1C). All populations were identified based on computed cluster-specific markers and well-established marker genes (Appendix A). Cell clusters were identified as keratinocytes, fibroblasts, immune cells, endothelial cells, muscle and smooth muscle cells and SEB. Interestingly, the relative numbers of the different cell types varied between the skin and PG (Figure 1D). Whereas PGs contained fewer keratinocytes, relative numbers of immune cells and SEB were higher in PGs. Although SEB clustered together in the skin and PGs (Figure 1C, purple), a striking difference in cluster shape was observed, already indicating differences in gene expression.

For a more detailed characterization, we subclustered SEB yielding five distinct SEB subpopulations in skin SGs and three in PGs (Figure 2A). Closer characterization of the subpopulations present in the skin showed high expression levels of the hair follicle (HF)-associated genes, such as *Krt27*, *Krt17*, *Hoxc13* and *Wnt5a* in the major three subpopulations (HF/SEB1, HF/SEB2, HF/SEB3) (Appendix A) [55,56,57,58,59,60,61]. The two main clusters in PG were identified as early SEB and late SEB and were both also found in the skin, though at significantly smaller numbers. The HF/SEB3 cluster was present in both skin and PG (Figure 2B). A total number of 663 genes was differentially expressed between sSEB and pSEB with a fold change cutoff greater than 1.5 or smaller than 0.6 (452 genes upregulated and 211 genes downregulated in the skin) (Figure 2C). In order to analyze transcriptional differences and, consequently, potential variations in the function, we performed gene ontology (GO) term enrichment analyses based on computed marker gene lists of sSEB and pSEB. sSEB GO terms were highly related to skin-specific functions such as epidermal development, epithelial cell differentiation, cell junction and cell–cell adhesion processes, as well as gland development (Figure 2D).

Contrary, genes highly expressed in pSEB were strongly related to lipid-specific GO terms involving lipid-binding and lipid-metabolic processes. Furthermore, genes involved in steroid metabolic processes were enriched in pSEB (Figure 2D). In addition, GO term analyses indicated that the sSEB cluster represented a mixture of lipid-specialized SEB and HF cells. Closer analysis of key genes associated with the previously identified GO terms showed that the HF/SEB3 in skin showed high expression levels of genes involved in skin-specific functions while HF/SEB3 in PG exhibited little to no expression of those genes but high expression of lipid-associated genes (Figure 2E). Together, our data suggest remarkable tissue-specific differences between sSEB and pSEB in gene expression and consequently indicate different SEB functions.

### 2.2. sSEB and pSEB Exhibit Significant Differences in Their Differentiation Program

As our previous results suggest a high tissue-specific gene expression profile, we first compared the differentiation programs engaged by sSEB and pSEB. Analyses of key genes involved in the regulation of sebaceous cell differentiation revealed that the HF-associated clusters, HF/SEB1, HF/SEB2 and HF/SEB3, as well as early SEB in the skin, exhibited high expression levels of β-catenin (*Ctnnb1*) (Figure 3A). In addition, the Wnt/β-catenin downstream effector gene *Lef1*, which promotes differentiation towards HF cells, was strongly expressed in HF/SEB1 and HF/SEB2 clusters (Figure 3B) [9,10]. In contrast, only early and late pSEB expressed high levels of β-catenin (*Ctnnb1*) but no *Lef1* (Figure 3A,B). This finding is concomitant with previously published studies describing canonical Wnt/β-catenin signaling is critical for cell fate decision towards either HF or SG in the skin [8,10,62]. Both early and late SEB in skin and PGs showed high expression levels of *Pparg* (Figure 3C) and *Plin2* (Figure 3D) compared to HF-associated clusters (Figure 3D), indicating an advanced SEB differentiation stage. We furthermore performed pseudotime trajectory analysis of skin HF-keratinocytes (HF-KC), identified by HF-associated genes (Appendix A) and sSEB (Figure 3E) [63]. We included HF-KCs as the HF-associated SEB cluster indicated a potential cell fate decision towards differentiated keratinocytes with lineage affinity towards hair follicle keratinocytes (HF/Diff KC). The starting point (green) was set at *Blimp1*^+^ (*Prdm1*) expressing stem cells (Figure 3E, Appendix A). The first branching point (white circle) after pseudotime start (green circle) directed cell differentiation either into SEB or HF cells. Within the HF/Diff KC cluster, there were two additional fate decision checkpoints, indicating differentiation steps within the HF cluster into different KC subpopulations found in HFs. Once the cell fate decision was made towards SEB, there was only one differentiation endpoint (marked as red circle). For pSEB, such trajectory analysis was not computed because stem cells or precursors could not be mapped with high accuracy (Appendix A). In summary, these findings clearly indicate that sSEB and pSEB undergo strikingly dissimilar differentiation programs to reach a lipid-specialized and late differentiated state.

### 2.3. Key Enzymes of Critical Lipid Synthesis Pathways Are Differentially Expressed in sSEB and pSEB

In order to investigate differences in the lipid synthesis pathways between sSEB and pSEB, we analyzed the expression of enzymes important for the production of squalene and sphingolipids.

Analysis of the expression levels of key enzymes involved in the synthesis of squalene (Figure 4A) revealed that most of these enzymes (*Hmgcs1*, *Hmgcr* and *Fdps)* were significantly increased in late SEB in both tissues (Figure 4B–D). Interestingly, expression levels of the squalene synthase (*Fdft1)* differed significantly between the skin and PGs (Figure 4E). Whereas *Fdft1* was highly expressed in sSEB, its expression was almost completely absent in pSEB, suggesting strongly reduced cholesterol synthesis. Together, these data indicate that pSEB-derived lipids produced via the mevalonate/isoprenoid pathway are more likely to follow the non-sterol branch.

Our scRNAseq data revealed significant differences in expression levels of genes involved in the synthesis of bioactive sphingolipids (Figure 5A). Figure 5B shows the average gene expression levels of key enzymes involved in de novo, sphingomyelinase and salvage lipid synthesis. *Sptlc1*, the key enzyme for de novo synthesis, was significantly higher expressed in pSEB compared to sSEB (Figure 5B,C). In contrast, sSEB showed higher expression levels of *Smpd1*, critically involved in ceramide synthesis from sphingomyelin (Figure 5B,C). Enzymes that further process ceramide into either sphingomyelin or other sphingolipids within the Golgi, such as *Sgms1* and *Ugcg*, were expressed at higher levels in sSEB, or were completely absent from pSEB (Figure 5B). Not only the initiating enzyme of sphingomyelin-ceramide conversion, *Smpd1*, but also downstream effector enzymes within the endo-lysosomal compartment, such as the acid ceramidase *Asah1,* exhibited higher expression levels in sSEB (Figure 5C). Furthermore, *Cers4* expression was found to be almost exclusively linked to sSEB, while pSEB showed higher specificity for *Sphk1* (Figure 5D). To validate our scRNAseq data, we performed immunofluorescent staining of some of the differentially expressed lipid-metabolizing key enzymes in the skin and PG (Figure 5E–G). While *Sptlc1* expression was strongly expressed in sebocytes of PG, SG showed only marginal staining (Figure 5E). In contrast, *Asah1* (Figure 5F) and *Cers4* (Figure 5G) protein expression were significantly higher in the SG as compared to PG (Figure 5F).

Together, these differences in enzyme expression levels suggest that pSEB sphingolipid synthesis occurs rather de novo with subsequent processing of ceramide to sphingosine in order to generate sphingosine-1-P. Contrary, sSEB appears to synthesize a broader range of bioactive sphingolipids and recycle free sphingosine to ceramide via the salvage pathway.

## 3. Discussion

Lipids are bioactive signaling molecules, triggering a vast array of cellular responses [30]. In the skin, SG are the main producers of lipids, specifically sphingolipids and the cholesterol precursor squalene, both of which critically contribute to the establishment and maintenance of the skin barrier [18,64,65,66,67,68]. Since PG have often been used as a model for skin SG, we analyzed both tissues on a single cell level and found striking differences in the differentiation program and lipid synthesis processes in sebocytes of both organs.

sSEB differentiation from sebocyte stem cells is described to follow a tightly controlled program that allows the cells to either differentiate into HF cells or SG cells [10]. By contrast, pSEB differentiation is only poorly described, and our data suggest that it follows a different developmental program. Our single-cell analysis identified the ability of sSEB to differentiate into HF cells as the major difference between both organs. Especially genes downstream of β-catenin were not expressed in pSEB [9,10]. Interestingly, *Blimp1^+^* SG stem cells were only identified in sSEB but not in pSEB, suggesting that the SEB stem cells already differ significantly between SG and PG. It is yet unclear which cells serve as pSEB precursors and which differentiation pathway is crucial for lipid-specialized cell fate decisions. Thus, pSEB cannot substitute sSEB in the in vitro studies investigating sSEB differentiation and potential ensuing consequences for HF homeostasis.

In addition to these fundamental dissimilarities, sSEB and pSEB further showed significant differences in expression levels of key enzymes involved in lipid synthesis. Squalene serves as an SG-derived precursor for cholesterol synthesis by keratinocytes [22]. Interestingly, the literature on squalene in rodent sebum and preputial glands is rather controversial. While some previous studies showed that the cholesterol precursor squalene is absent from rodent skin surface lipids and preputial glands, others identified squalene in both organs [69,70,71,72,73]. It is noteworthy that these studies, providing the foundation for modern research, relied on volatile analysis methods [70,71]. As cholesterol is found on the skin surface and within preputial gland lipids and squalene–cholesterol conversion occurs at rather high frequency with low storage rates of squalene in rodents; the majority of reports indicate that squalene is indeed produced in both organs [24,70,71,72,74,75]. In addition, we found high levels of key enzymes of the mevalonate pathway in sSEB and pSEB, which is upstream of squalene synthase. Interestingly, we detected remarkable differences in the expression of squalene synthase between sSEB and pSEB. The low levels of squalene synthase found in pSEB suggest that these cells rather produce non-sterol compounds, while the synthesis of squalene in sSEB offers a decision between the non-sterol and the sterol branch, eventually leading to cholesterol synthesis by keratinocytes [76]. Given that cholesterol can either be synthesized de novo via the mevalonate pathway, or taken up from peripheral tissues via binding of low-density lipoprotein (LDL) and high-density lipoprotein (HDL), pSEB may not require squalene as cholesterol precursor and circumvent squalene synthase by LDL or HDL receptor-mediated endocytosis [27,76]. This substitute source of cholesterol may contribute to sufficient levels of cholesterol independent of de novo synthesis for pheromone-associated functions in the PG. Of note, besides serving as a cholesterol precursor, squalene is also known to protect the skin from UV-irradiation-induced injury [77]. In addition, UV irradiation triggers a decomposition of skin surface lipids, specifically squalene, resulting in the generation of lipoperoxides of squalene, which are able to reduce contact hypersensitivity in mice [78]. Since PG are not reached by UV radiation, this protective mechanism is irrelevant in PG and may result in a generally reduced requirement for squalene synthesis.

Previous studies implicated CER in intracellular as well as stress-induced signaling responses influencing keratinocyte proliferation, differentiation and apoptosis [79]. In terms of maintaining epidermal barrier function, ceramide is believed to act on PPAR, which stimulates keratinocyte differentiation [40]. Furthermore, PPAR activation leads to ATP binding cassette transporter family-12 production serving as a critical regulator of glucosylceramide delivery to lamellar bodies, which in turn contributes to epidermal barrier formation [80,81]. Considering the high potency of ceramide as the second messenger in intracellular signaling critically influencing epidermal barrier function, we investigated key enzymes involved in de novo as well as salvaged sphingolipid synthesis [18,80,81,82,83,84]. Our scRNA sequencing data revealed that rate-limiting enzymes in sphingolipid synthesis, such as serine palmitoyltransferase (*Sptlc1*) and sphingomyelinase (SMase; *Smpd1*), are present in SEB of both tissues. However, our data suggest that sphingolipid synthesis in PGs favors de novo synthesis, as *Sptlc1* expression levels were significantly higher in pSEB. The most prominent difference was identified in the salvage pathway, as pSEB showed little to no ceramide synthase (*CerS*) expression but high expression of sphingosine kinase 1 (*Sphk1*). Together, these findings indicate that whereas sSEB preferentially produces CER, pSEB favors the production of sphingosine-1-phosphate (S1P) from sphingosine. Besides the crucial contribution of ceramide to the establishment of a functional skin barrier, it was previously linked to suppressing steroid hormone production, which is a major function of the PG [85,86,87,88,89,90,91]. In contrast, *Sphk1* and its lipid product S1P were linked to promoting steroid hormone production [88,92,93,94,95,96]. Taken together, our data suggest a higher affinity of sSEB to produce ceramide, independently of either the de novo or salvage route, while pSEB appear to be more specialized in lipid production relevant for steroid hormone production, where increasing ceramide levels are unfavorable. Additionally, ceramide metabolites such as glucosylceramide and glycosphingolipids are of vital importance for epidermal barrier function by either serving as precursors for stratum corneum derived ceramide or actively contributing to skin barrier homeostasis [95,96]. The discrepancy in enzymatic expression levels important for the conversion of ceramide to glucosylceramides and glycosphingolipids further indicates fundamental differences between sSEB and pSEB.

This study has several limitations. The main limitation of this study is the fact that we solely investigated male murine SEB populations of different tissue origins, which are not fully comparable with human data. Furthermore, we focused our investigations on the expression of lipid metabolizing enzymes without further analysis of lipid production. Comparably detailed analyses of the lipid metabolizing enzymes investigated in this study have not yet been conducted in human SG. As other publications showed significant differences in sebum composition of humans and rodents [24], it is likely that significant differences in metabolizing enzymes are also present between those species. In summary, besides superficial transcriptional similarities and morphological resemblances, pSEB shows tremendous differences rendering them highly unfavorable as a substitute model for sSEB. Due to their different physiological properties (i.e., contribution to skin homeostasis and epidermal barrier function versus reproduction and sexual behavior of rodents [1,97]), alterations in the expression of lipid-modifying enzymes were, however, anticipated. Our study has built the basement for further studies, which will be necessary to fully understand the fundamental differences in the various SEB populations and to elucidate which results obtained from pSEB can indeed be extrapolated to sSEB.

## 4. Materials and Methods

### 4.1. Animals

Three male wild type C57BL/6 mice at the age of 52 weeks were used for this study. Biopsies were obtained from back skin and preputial glands. Skin biopsies, as well as preputial gland samples, were pooled respectively and subsequently processed for single-cell RNA sequencing, protein lysates or histology.

### 4.2. Single Cell RNA Sequencing (scRNAseq)

Immediately after obtaining the skin biopsies and preputial glands, tissues were enzymatically digested using GentleMACS whole skin dissociation kit (Miltenyi Biotec, Bergisch-Gladbach, Germany) according to the manufacturer’s protocol. Samples were further processed on a GentleMACS OctoDissociator (Miltenyi). The cell suspension was filtered through 100 µm and 40 µm filters and washed with 0.04% bovine serum albumin (BSA, Sigma Aldrich, St. Louis, Mo, USA) in PBS. Cell viability and concentration was assessed using Acridine Orange/Propidium Iodide (AO/PI) Cell Viability Kit (Logos Biosystems, Anyang-si, Gyenoggi-do, Korea) and analyzed on a LUNA-FLDual Fluorescence Cell Counter (Logos Biosystems). Cell concentrations were adjusted to 0.7 to 1.2 × 10^6^ cells/mL. Subsequently, gel beads-in-emulsion (GEMs) were generated. GEM generation, barcoding, sample clean-up, cDNA amplification as well as library construction were performed in accordance with the manufacturer’s protocol using Chromium Next GEM Single Cell 3′GEM, Library and Gel Bead Kit v3.1, Chromium Next GEM Chip G Single Cell Kit and Single Index Kit T Set A (10x Genomics, Pleasanton, CA, USA). Final RNA-sequencing, demultiplexing and counting were performed by the Biomedical Sequencing Facility (BSF) of the Center for Molecular Medicine (CeMM, Vienna, Austria). Obtaining samples and acquiring cell suspensions by Chromium instrument occurred within 4 h. In total, data of approximately 10,000 cells were obtained.

### 4.3. Data Analysis (Bioinformatics)

Bioinformatics analyses were carried out using R (R version 4.0.3, The R Foundation, Vienna, Austria), R-studio desktop application and the R-Package “Seurat” (Seurat v4.0.3, Satija Lab) [98,99,100,101]. To reduce variations, cells with very low or very high unique molecular identifiers (UMI) or a high percentage of mitochondrial gene counts were excluded for subsequent analyses. PCA and UMAP were calculated, and all further subset analyses were based on raw data of the cells of interest. Different cell populations were identified based on clustermarker features and checked for well-established marker genes for sophisticated cluster definition (Appendix A). Gene ontology (GO) enrichment analyses were carried out based on clustermarker and differentially expressed gene calculations. Only genes with a fold change greater than 1.5 or smaller than 0.6 were included. The publicly available enrichment tool Metascape was used for GO enrichment analyses [102]. A *p*-value cutoff of 0.01 and a minimum enrichment score of 3 were set for GO terms with molecular function and biological process membership.

### 4.4. Immunofluorescence and H&E

For immunofluorescent staining, tissue samples were formalin-fixed, embedded in paraffin and cut into 5 µm thick sections. Paraffin sections were deparaffinized by heating and ethanol dilution series (100%, 80%, 30%). Antigen de-masking was performed with Dako TRS citrate buffer at pH6 (Agilent Technologies, Santa Clara, CA, USA). Sample sections were incubated with anti *Scd1* (ab236868, Abcam, Cambridge, UK), anti *Asah1* (ab282276, Abcam), anti *Sptlc1* (ABIN635127, antibodies-online, Aachen, Germany) or *Cers4* (ABIN2434915, antibodies-online) primary antibody diluted in 2% BSA in PBS overnight at 4 °C. The secondary antibody was diluted in 2%BSA in PBS. Hematoxylin and eosin staining was performed according to routine laboratory procedures. Images were acquired with an Olympus BX63 microscope (Olympus, Tokyo, Japan) with MetaMorph imaging software.

## Figures and Tables

**Figure 1 ijms-22-11631-f001:**
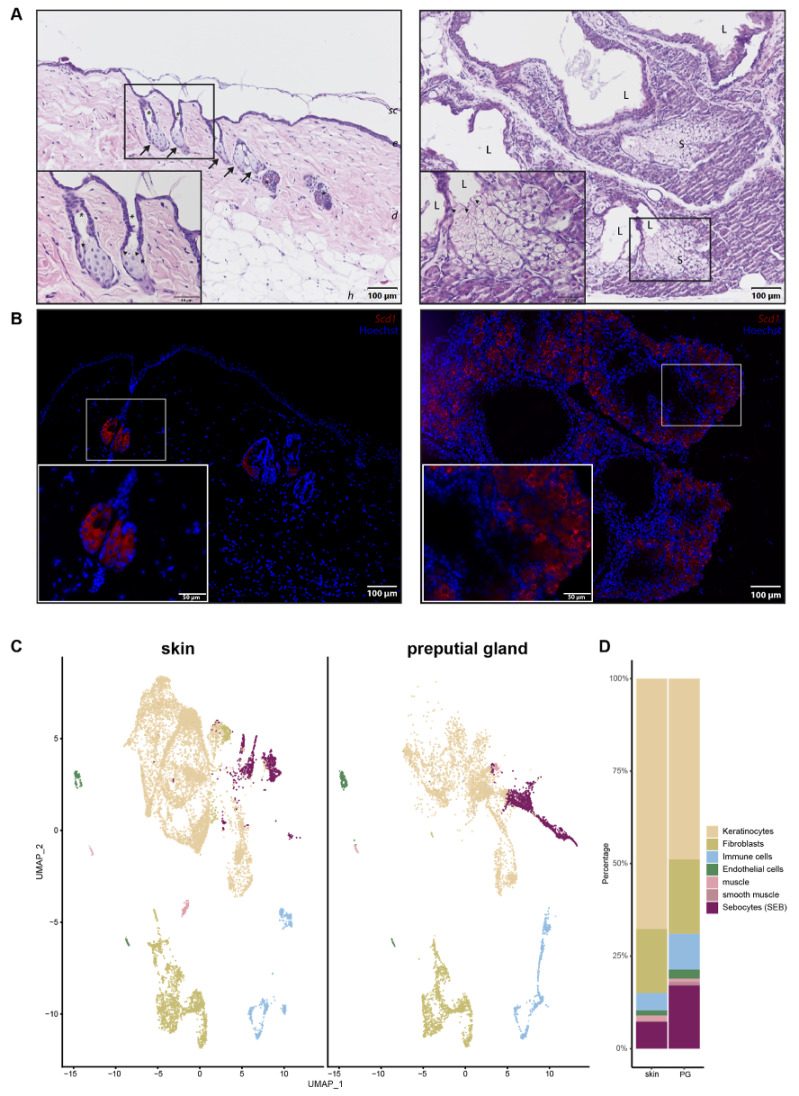
Comparison of skin and preputial gland sebocytes. (**A**) Hematoxylin and eosin staining of a skin biopsy (left panel) and a preputial gland (right panel). Arrows indicate sebaceous glands in close proximity to hair shafts marked by asterisks. Arrowheads indicate differentiated sebocytes with degenerated nuclei, facing the lumen of either the hair shaft or the lumen of the preputial gland; Scale bar = 100 µm, 50 µm in insert; sc: stratum corneum; e: epidermis; d: dermis; h: hypodermis; L: lumen of the preputial gland; S: sebocytes; (**B**) Representative immunofluorescence labeling of *Scd1^+^* sebocytes in the skin (left panel) and the preputial gland (right panel); scale bar = 100 µm, 50 µm in insert. Tissues of *n* = 3, respectively. (**C**) UMAP-plots comparing cells of skin biopsies (*n* = 3) and preputial glands (*n* = 3), split by tissue, identifying keratinocytes, fibroblasts, immune cells, endothelial cells, muscle and smooth muscle cells and sebocytes (SEB). (**D**) Bar plot indicating relative numbers of cell populations identified within the skin and preputial gland.

**Figure 2 ijms-22-11631-f002:**
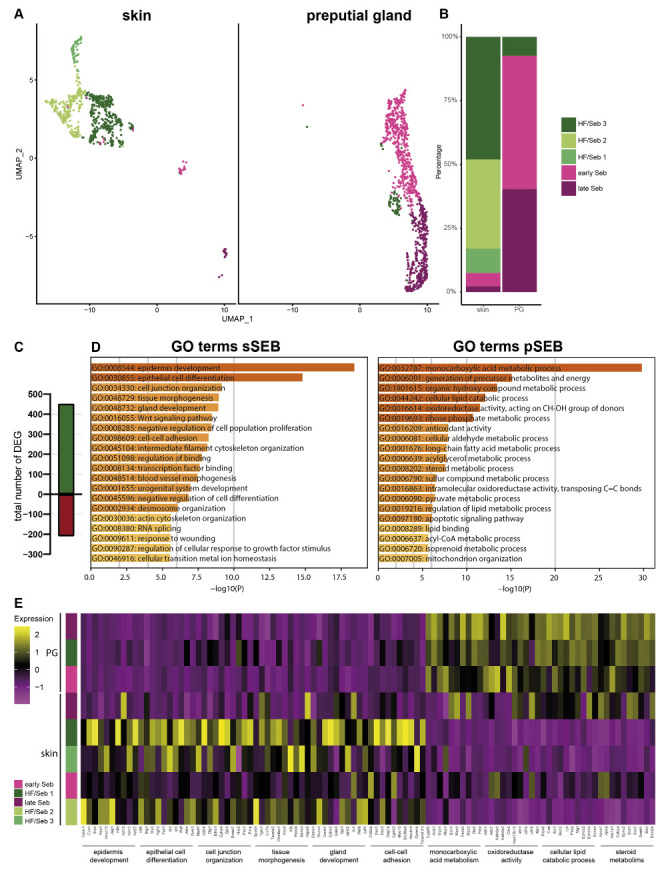
Subclustering of sebocytes and Gene Ontology analysis (**A**) Analysis of the sebocyte subcluster identified five distinct subtypes in UMAP-plot. Subclusters were identified as HF/SEB 1, HF/SEB 2, HF/SEB 3, early SEB and late SEB. (**B**) Barplot indicates relative cell numbers present within each sebocyte subcluster. (**C**) Barplot shows the total number of differentially expressed genes of total sSEB compared to total pSEB. (**D**) Gene Ontoloy (GO) term analysis results using Metascape. Genes with a differential expression fold change of >1.5 or <0.6 were assigned to GO terms. (**E**) Heatmap showing expression levels of genes associated with distinct GO terms in the different SEB subclusters.

**Figure 3 ijms-22-11631-f003:**
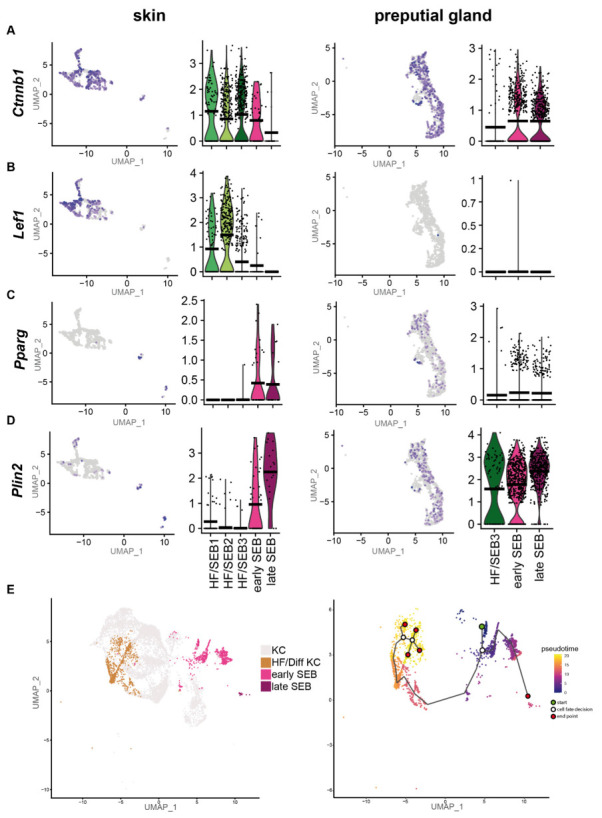
Preputial gland sebocytes lack downstream effector genes relevant for HF/SG lineage decisions. Featureplots and violin plots of the integrated sebocyte subcluster from skin and preputial gland showing the expression of (**A**) β-catenin (*Ctnnb1*), (**B**) Lymphoid enhancer-binding factor-1 (*Lef1*), (**C**) Peroxisome proliferator-activated receptor gamma (*Pparg*), (**D**) Perilipin 2 (*Plin2*). Violin plots show gene expression levels and crossbar of violin plots depicts mean expression value. Vertical lines show maximum expression. The width of violins represents the frequency of cells at the corresponding expression levels. (**E**) UMAP-plot and pseudotime trajectory track starting from *Blimp*^+^ sebocyte precursors (green circle). Color code indicates computational calculated progression in differentiation. White circles indicate cell fate decision checkpoints and red circles depict differentiation endpoints.

**Figure 4 ijms-22-11631-f004:**
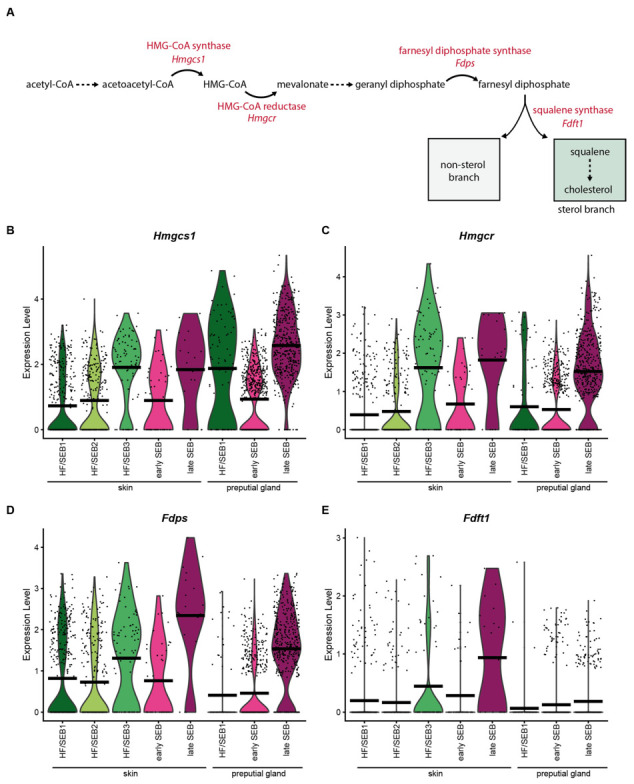
Key enzymes of the mevalonate lipid synthesis pathway are less expressed in preputial gland sebocytes. (**A**) Schematic depiction of the mevalonate pathway and key enzymes involved in the synthesis of squalene. (**B**–**E**) Violin plots showing the expression level of (**B**) HMG-CoA-synthase (*Hmgcs1*), (**C**) HMG-CoA reductase (*Hmgcr*), (**D**) farnesyl diphosphate synthase (*Fdps*), and (**E**) squalene synthase (*Fdft1*). Violin plots show gene expression levels and crossbar of violin plots depicts mean expression value. Vertical lines show maximum expression. Width of violins represents the frequency of cells at the corresponding expression levels.

**Figure 5 ijms-22-11631-f005:**
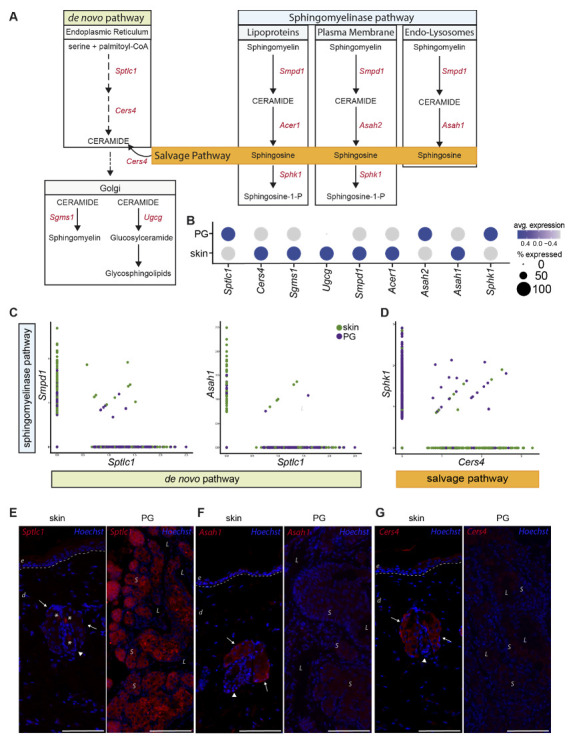
Preputial gland sebocytes are limited to de novo biosynthesis of sphingolipids. (**A**) Schematic drawing of the de novo, sphingomyelinase and salvage pathway of ceramide- and ceramide metabolite synthesis. Key enzymes involved in the respective pathways are colored in red. *Sptlc1*: serine palmitoyltransferase1; *Cers4*: ceramide synthase 4; *Sgms1*: sphingomyelin synthase 1; *Ugcg*: ceramide glucosyltransferase; *Smpd1*: sphingomyelinase 1; *Acer1*: alkaline ceramidase; *Asah2:* neutral ceramidase; *Asah1*: acid ceramidase; *Sphk1*: sphingosine kinase 1. (**B**) Dot plot of average gene expression of the key enzymes involved in the synthesis of sphingolipids in preputial gland and skin tissue. Point size indicates relative averaged amount of positive cells. Color code indicates average gene expression levels. (**C**) Featurescatter-plot of sSEB and pSEB shows correlation gene expression of *Smpd1* and *Sptlc1*, or *Asah1* and *Sptlc1*. (**D**) Featurescatter-plot of sSEB and pSEB showing correlational gene expression of *Sphk1* and *Cers4*. Gene expression levels for each gene, respectively, is shown on either x- or *y*-axis. Immunofluorescent labeling of skin or preputial gland of (**E**) *Sptlc1*, (**F**) *Asah1* and (**G**) *Cers4*. Scale bars = 100 µm. Arrows indicate sebaceous glands in close proximity to hair shafts marked by arrowheads. Asteriscs in (**E**) left panel indicate staining of cell-free material; e: epidermis; d: dermis; L: lumen of the preputial gland; S: sebocytes.

## Data Availability

ScRNASeq data are available in NCBI’s Gene Expression Omnibus (GEO); accession number GSE185268.

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
