# Peer review of "Transcriptional Differences in Lipid-Metabolizing Enzymes in Murine Sebocytes Derived from Sebaceous Glands of the Skin and Preputial Glands"

_ijms, 2021, doi:10.3390/ijms222111631_

Round 1
Reviewer 1 Report
The paper describes differences between mouse SG and PG in terms of differentiation route and lipid production based on scRNAseq. The approach takes advantage of that scRNAseq can provide cell type specific transcriptome-wide data from a limited amount of tissue consisting of mixed cell types. A comparable differentiation program and lipid production of sebocytes are crucial aspects for investigators in choosing their experimental model.
My minor suggestion/comment:
The conclusions are exclusively based on differential gene expression analysis of scRNAseq data. The results are interesting but should be confirmed by IHC for the identified differentially expressed genes.
Author Response
Point to point reply
We thank the editor for the interest in our work and the reviewers for their effort and time put into the thoughtful review and their comments. Each comment has been carefully considered point by point. To facilitate the work of the reviewers, the new parts in the manuscript are marked up using the “Track Changes” function.
Comment 1
The conclusions are exclusively based on differential gene expression analysis of scRNAseq data. The results are interesting but should be confirmed by IHC for the identified differentially expressed genes.
Reply to comment 1
We have primarily conducted our study to investigate differences in gene expression of murine sebocytes derived from skin and preputial glands on a single cell level. However, we agree with the reviewer that confirming our data by immuno stainings would strengthen our findings. According to the reviewer´s suggestion we have now included immunofluorescence stainings for Asah1, Sptlc1 and Cers4 in the revised version of our manuscript. The new stainings are shown in Figure 5E and demonstrate that protein expression of these enzymes is indeed comparable to the observed gene expression levels.
Reviewer 2 Report
This study demonstrated transcriptional differences of by lipid-metabolizing enzymes in skin and preputial gland sebocytes in mice by the single cell RNA sequencing analysis, suggesting that these 2 types seem to play the different roles, respectively: skin sebocytes tend to produce ceramide, while preputial sebocytes appear to be specialized in lipid production. The authors concluded that preputial sebocytes have tremendous differences rendering them highly unfavorable as substitute model for skin sebocytes.
This is well described. I think that it was firstly and extensively investigated using a novel technology for transcriptional analysis for lipid-metabolizing enzymes, although these data were confined to mice, which seemed different from those of humans.
There are several comments:
- This title does not contain “murine study”, which may confuse the readers. Please insert the murine study in the title.
- In the second paragraph in the Introduction section, it might be difficult to understand. Please explain it using the Figure 4A.
- Please discuss about physiological roles of sebocytes from skin and preputial glands, considering the difference in transcriptional expression of lipid-metabolizing enzymes. Can these results apply to humans? Do the authors have any speculations for humans?
Author Response
Point to point reply
We thank the editor for the interest in our work and the reviewers for their effort and time put into the thoughtful review and their comments. Each comment has been carefully considered point by point. To facilitate the work of the reviewers, the new parts in the manuscript are marked up using the “Track Changes” function.
Comment 1
This title does not contain “murine study”, which may confuse the readers. Please insert the murine study in the title.
Reply to comment 1
We agree with the reviewer that our title might be misleading. We have therefore changed the title in the revised version of our manuscript accordingly. The new title reads: “Transcriptional differences of lipid-metabolizing enzymes in murine sebocytes derived from sebaceous glands of the skin and preputial glands.”
Comment 2
In the second paragraph in the Introduction section, it might be difficult to understand. Please explain it using the Figure 4A.
Reply to comment 2
According to the reviewer´s suggestion we have rephrased the second paragraph of our introduction.
Comment 3
Please discuss about physiological roles of sebocytes from skin and preputial glands, considering the difference in transcriptional expression of lipid-metabolizing enzymes. Can these results apply to humans? Do the authors have any speculations for humans?
Reply to comment 3
The reviewer has raised an important point. Indeed, the physiological differences between both glands have been already discussed in our initial submission, by comparing epidermal barrier specific mechanisms and pheromone and steroid hormone production. To further point this out, we have now added a conclusive statement summarizing this topic in the discussion section of our revised manuscript (page 14). In addition, we have now added a paragraph discussing similarities and differences between skin sebocytes in mice and humans in the discussion section (page 14).